# Endocrine Management of Transgender Adults: A Clinical Approach

Stefano Iuliano [1,†], Giulia Izzo [1,†], Maria Carmela Zagari [1], Margherita Vergine [1], Francesco Saverio Brunetti [2], Antonio Brunetti [2], Luigi Di Luigi [3] and Antonio Aversa [1,*]

1   Department of Experimental and Clinical Medicine, Magna Graecia University of Catanzaro, 88100 Catanzaro, Italy; stefano.iuliano@studenti.unicz.it (S.I.); giulia_izzo@unicz.it (G.I.); maricazagari@gmail.com (M.C.Z.); margherita.vergine@hotmail.it (M.V.)
2   Department of Health Sciences, Magna Graecia University of Catanzaro, 88100 Catanzaro, Italy; bfr333@gmail.com (F.S.B.); brunetti@unicz.it (A.B.)
3   Unit of Endocrinology, Department of Movement, Human and Health Sciences, University of Rome "Foro Italico", 00135 Rome, Italy; luigi.diluigi@uniroma4.it
*   Correspondence: aversa@unicz.it
†   Equal contribution.

**Abstract:** The attention to transgender medicine has changed over the last decade and the interest is most likely going to increase in the future due to the fact that gender-affirming treatments are now being requested by an increasing number of transgender people. Even if gender-affirming hormone therapy (GAHT) is based on a multidisciplinary approach, this review is going to focus on the procedures adopted by the endocrinologist in an out-clinic setting once an adult patient is referred by another specialist for 'gender affirming' therapy. Before commencing this latter treatment, several background information on unmet needs regarding medical and surgical outcomes should be investigated. We summarized our endocrinological clinical and therapeutic approaches to adult transgender individuals before and during GAHT based on a non-systematic review. Moreover, the possible relationships between GAHT, gender-related pharmacology, and COVID-19 are also reported.

**Keywords:** transgender; healthcare; hormonal treatment; follow-up; endocrinologist

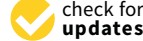



## 1. Introduction

Over the past 10 years, awareness about transgender medicine has increased. Gender identity refers to "a person's deeply-felt, inherent sense of being a boy, a man, or a male, a girl, a woman, or a female, or an alternative gender (e.g., genderqueer, gender nonconforming, gender neutral) that may or may not correspond to a person's gender assigned at birth or to a person's primary or secondary sex characteristics" [1]. In this narrative review, we shall focus our attention on the endocrine clinical and therapeutic approach of people experiencing Gender Dysphoria (GD) as defined by a distress due to "a marked incongruence between a person's gender assigned at birth and gender identity" [2].

It has been estimated a GD prevalence rate of 0.005–0.014% for assigned male at birth (AMAB) and 0.002–0.003% for assigned female at birth (AFAB) people [3].

In addition to binary GD, health-care professionals (HCPs) working with transgende people should also recognize individuals who do not identify themselves exclusively as either men or women (i.e., non-binary individuals). Patients asking for a gender-affirming medical approach should be evaluated by HCPs with expertise in mental health to also identify the need of a mental health professional's intervention [4].

GD should be managed by a multidisciplinary team and diagnosis should be done by an expert physician with specific skills and abilities [5]. Based on fulfillment of diagnostic criteria by prior HCP's evaluations, we aimed to describe the role of the endocrinologist in the management of the different gender-affirming treatment options and of clinical

monitoring in transgender adults in our multidisciplinary single-center medical clinic dedicated to outpatients. In fact, also during the diagnostic workup, the endocrinologist is required to investigate on possible medical contraindications to the Gender Affirming Hormonal Treatment (GAHT). When a specific hormonal treatment for GD is practicable, it can be started without further investigations [6] after informing each patient about efficacy and possible side effects of GAHT and fertility preservation techniques [7]. The distress of living in an incongruent body and gender role is often present since childhood and adolescence, but the treatment of youths with gender incongruence is not a matter of discussion of the present review. Our general diagnostic-therapeutic approach is depicted in Figure 1.

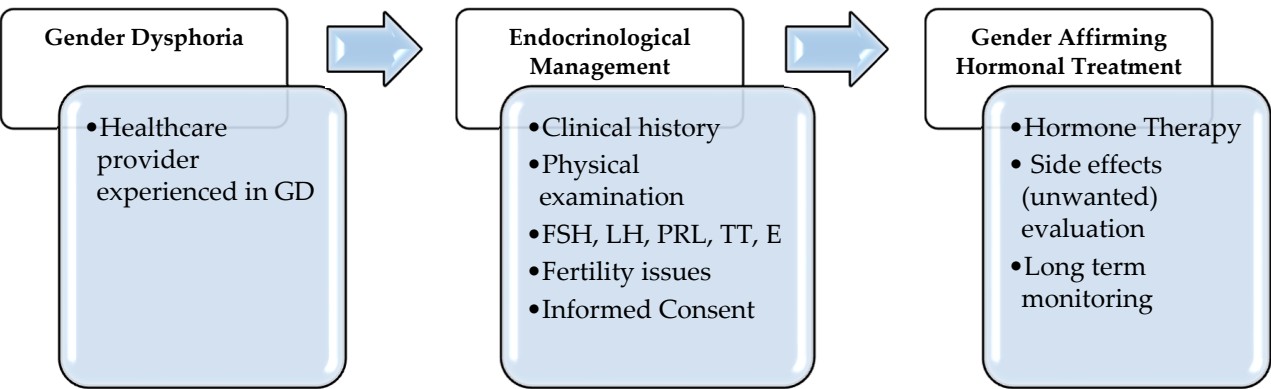

**Figure 1.** Diagnostic-therapeutic algorithm in Gender Dysphoria (GD). Legends: FSH = Follicle-Stimulating Hormone; LH = Luteinizing Hormone; PRL = Prolactin; TT = total testosterone; E = estradiol.

## 2. The Endocrinological Management of Transgender Men

In AFAB individuals experiencing GD and asking for full virilization, the main targets of GAHT (Table 1) are to suppress endogenous female hormones secretion, to reduce female secondary sexual characteristics, and to induce virilization, to satisfy patient's gender identification.

Our endocrinological approach in treating transgender men is depicted in Figure 2. Before starting hormonal therapy, particularly the patient's-oriented contraindications for testosterone administration must be assessed. The endocrinologist should also evaluate the presence/absence of other risk factors, such as alcoholism, smoking, chronic use of drugs, and intercurrent illness [8].

Before starting hormonal therapy, it is also necessary to discuss fertility options. To preserve fertility in transgender men there are some options available: Oocytes, embryo, and ovarian tissue banking [9]. Oocyte cryopreservation does not immediately require a partner or a sperm donor because there is no fertilization before banking. Embryo cryopreservation after hormonal stimulation and oocyte aspiration offer to AFAB individuals the possibility of a genetically related child with male partner. A sperm donor could be employed to create embryos; in addition, if the uterus is removed in transgender men undergoing gender affirming surgery, a recipient uterus is required (i.e., surrogate mother). Ovarian tissue banking needs a surgical procedure that could be performed during gender affirming surgery [10].

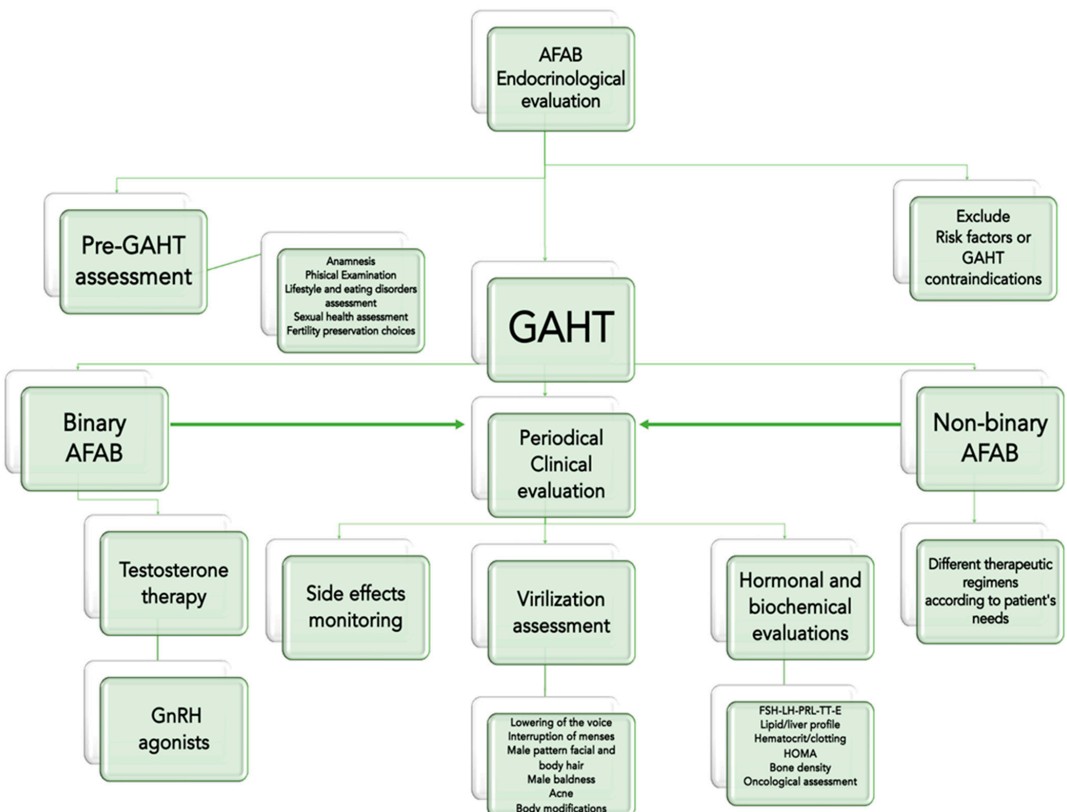

**Figure 2.** Clinical approach to individuals assigned as female at birth (AFAB) and affected by Gender Dysphoria, before and after a Gender Affirming Hormonal Treatment (GAHT). Legends: FSH = Follicle-Stimulating Hormone; LH = Luteinizing Hormone; PRL = Prolactin; TT = total testosterone; E = estradiol; HOMA = Homeostasis Model Assessment for insulin resistance (index).

**Table 1.** Drugs used for Gender Affirming Hormonal Treatment (GAHT) in transgender men. From [6], modified.

| Class | Drug | Administration Route | Dose | Risk Factors for Adverse Outcomes [11] | Side Effects |
|-------|------|---------------------|------|----------------------------------------|--------------|
| *Androgens* | *Testosterone undecanoate* | Oral | 160–240 mg/die | **Very high**: Breast/uterine cancer, polycythemia, venous thromboembolism **Moderate**: baseline hematocrit > 50%, uncontrolled congestive heart failure, untreated Obstructive Sleep Apnoea Syndrome | Hypertension, acute hepatitis, erythrocytosis, hydrosaline retention/edema, increased libido, psychiatric disturbances (in predisposed individuals) |
| | *Testosterone enanthate or cypionate* | Intramuscular | 100–200 mg/2 weeks | | |
| | *Testosterone undecanoate* | Intramuscular | 1000 mg/12 weeks | | |
| | *Testosterone gel* | Transdermal | 40–80 mg/die | | |
| | *Testosterone patch* | | 2.5–7.5 mg/die | | |
| *GnRH agonists* | *Leuprolide acetate* | Subcutaneous | 3.75 mg/30 days | **Not Reported** | Female hormones deprivation Sexual disturbances |
| | *Triptorelin* | | 11.25 mg/90 days | | |
| | *Goserelin* | | 3.6 mg/30 days | | |
| | | | 10.8 mg/90 days | | |

*2.1. GAHT Regimens and Side Effects*

If full virilization is required, the therapeutic regimen with androgens in GAHT does not differ from cisgender men affected by hypogonadism [12] and many testosterone preparations can be used; in our clinical practice, we prefer either parenteral or transdermal (i.e., gel) testosterone formulations (Table 1). Moreover, in agreement with clinical judgement, progestin or gonadotropin-releasing hormone (GnRH) agonists could be prescribed

in selected cases to stop menses, according to the patient's wishes and standard safety procedures (Table 1). Generally, GAHT regimens should be considered effective and at low risk for serious side effects [13], even if patients may sometimes develop erythrocytosis (Hct > 50%, estimated incidence rate of 9.2%) [14], hair loss [15] and acne (within the first 6–9 months of treatment) [16].

Actually, the classic view of gender binarism and GAHT should be reconsidered because of the increasing prevalence of non-binary individuals experiencing GD. Therefore, new therapeutic strategies should be also considered [17]. Since there is a lack in standardized hormonal treatment protocols for non-binary AFAB, GAHT goals should be adjusted according to patients' needs in order to improve the self-perception and the quality of life [18]. For instance, different testosterone doses and/or other androgen preparations (i.e., nandrolone) are suggested to modulate the requested body changes [18], even if serious ethical concerns on some drugs exist.

The first physical changes are visible after 1–3 months of GAHT (e.g., body fat redistribution [13] and increase in Body Mass Index (BMI)), while many others such as increase in lean mass [8] or deepening of voice need more time to be noticeable [19] and may become appreciable after 1–2 years. Velho et al. reported a correlation between increased BMI and significant lean mass gain [20].

The effects of GAHT on sexual functioning are unclear since evidence is still scant. However, an increase in sexual desire, sexual arousal, and satisfaction have been described during the testosterone treatment follow up [4].

## 2.2. Clinical, Hormonal and Biochemical Evaluations

After starting therapy, transgender men must undergo periodical clinical checkups to assess treatment's efficacy and evaluate unwanted side effects (Table 1).

During the first year of GAHT, a tri-monthly clinical monitoring is recommended; and subsequently every six months or annually. At each clinical evaluation during GAHT, several parameters such as weight, height, handgrip, and chest circumference should be evaluated. Furthermore, Ferriman–Gallwey score and Global Acne Grading Score can be also used to estimate hair and acne clinical variations [6,8].

In addition to physical examination monitoring, hormonal and biochemical parameters are also necessary at the same timepoints (i.e., total testosterone, luteinizing hormone (LH), hematocrit, lipid profile, liver function tests (e.g., alanine aminotransferase, aspartate aminotransferase, alkaline phosphatase, and so forth) [6].

To correctly interpret hormonal and biochemical parameters, clinicians should be familiar with new appropriate reference ranges. However these latter still need international validation [21].

Serum total testosterone concentration should be maintained in the male's normal range, between 350 and 750 ng/dL [12]. In addition to the virilization process, adequate serum testosterone levels are necessary to maintain bone mineral density (BMD) in transgender men. Indeed, an increase in LH levels suggests an inadequate hormonal therapy and inversely relate with BMD [22].

Regarding lipid profile modifications, in patients who underwent hormonal treatment with testosterone, metanalysis did not provide robust evidence. However, an increase in triglycerides [23], low-density lipoprotein (LDL) cholesterol [4], and total cholesterol [20] has been reported. On the contrary, high-density lipoprotein (HDL) cholesterol is reduced by GAHT [4].

There is no consensus about possible relationships between GAHT and metabolic syndrome in transgender men. It is known from metanalytic studies that testosterone replacement therapy (TRT) is associated with an improvement in glucometabolic parameters, such as fasting plasma glucose and homeostatic model assessment (HOMA) index, and a reduction in waist circumference [24]. Evidence reported no relationship with insulin-resistance after 1 year of testosterone undecanoate in transgenders individuals [23,25]. Shadid et al. reported that incretin responses and insulin sensitivity parameters are im-

proved because of masculinization body composition shifts (i.e., increased lean mass and reduced fat mass) more than the weight changes [26].

Even if serious cardiovascular events have not been reported [27], there is a lack of evidence concerning the risks of stroke or myocardial infarction in transgender men [23]. Therefore, clinicians should manage these complications following current guidelines for the general population [6]. As a side note, the increase in hemoglobin and hematocrit observed in some studies could be a risk factor for patients at higher risk for thrombosis such as hypertensive or obese individuals [20].

Finally, in the absence of gender affirming surgery, periodical screening for uterus and breast cancer should be performed as in cisgender women [9].

### 2.3. Psychological Aspects in Transgender Men during Gender Affirming Hormonal Treatment (GAHT)

Individuals' gender identity is affected by different factors such as ethnicity, nationality, religion, profession, and age. Gender identity can be considered as one of the most important elements of human life [28]. Due to the irresistible urge of living in the other gender, transgender individuals undergo hormone therapy in order to stimulate the expression of their experienced gender. In individuals with GD asking for GAHT there is a higher prevalence of mental health disorders (i.e., disproportionate rates of stressful life events due to their GD that can lead to major psychiatric disorders such as anxiety or substances abuse) [29]. Personality and affective disorders are also described in this population [30].

Discrimination and social stigma are often experienced by transgender individuals. In a study of 402 transgender persons, more than half reported verbal harassments and more than a third reported employment discrimination. In accordance with these assessments, Jones et al. reported that transgender men are more likely to be diagnosed with depression [31]. Discrimination and non-affirmation can also induce suicidal ideation [32].

A Taiwanese study found a strong correlation between GD and schizoid personality in both men and women [33], but transgender men undergoing GAHT show a reduction in the paranoia subscale of the Minnesota Multiphasic Personality Inventory (MMPI) [34]. This means that, even though schizophrenia traits may be more likely to be diagnosed in people affected by GD, GAHT seems to reduce this association [35]. Androgens therapy seems to improve the quality of life and it also seems to reduce anxiety and social distress [36]. Even though the underlying mechanisms are still debated, untreated patients seem to suffer from a higher degree of stress compared to treated patients [37]. However, whether treatment is associated with a better mental well-being is still under debate [36]. According to Fisher et al., GAHT partially improves self-perception because of body modifications, such as increase in body mass index, that determine a psychopathology reduction [38].

Current studies have shown that in transgender individuals there is also an increased risk for eating disorders and an early diagnosis is critically important [39]. Silverstein et al. reported a higher prevalence of purging or frequent binging in AFAB who reported a gender identity conflict [40], while Ålgars et al. found that most transgender individuals experienced disordered eating trying to enhance sexual characteristics of their gender identity [41]. A recent study reported that body uneasiness may be the link between GD and eating disorders [42].

These results demonstrate the vulnerability of these individuals and the need for an awareness program for professionals [43].

### 3. The Endocrinological Management of Transgender Women

In transgender women, the main targets of GAHT (Table 2) are to suppress male secondary sexual characteristics and to enhance feminine ones to satisfy patient's expectations.

**Table 2.** Drugs used for Gender Affirming Hormonal Treatment (GAHT) in transgender women. From [6], modified.

| Class | Drug | Administration Route | Dose | Risk Factors for Adverse Outcomes | Side Effects |
|---|---|---|---|---|---|
| *Estrogens* | *Estradiol* | Oral | 2–6 mg/die | **Very high**: venous thromboembolism, prostate cancer **Moderate**: macroprolactinoma, breast cancer, coronary artery diseases, cerebrovascular diseases, cholelithiasis and hypertriglyceridemia. | Increase of thrombotic risk and hepatotoxicity |
| | *Estradiol valerate* | | | | |
| | *Estradiol valerate or cypionate* | Intramuscular | 2–10 mg/week 5–30 mg/2 weeks | | |
| | *Estradiol patch* | Transdermal | 0.025–0.2 mg/die | | Increase of thrombotic risk, hepatotoxicity and cutaneous reactions |
| *Antiandrogen-Progestin* | Cyproterone acetate | Oral | 25–50 mg/die | **Not Reported** | Increased risk of meningiomas, depression, and hyperprolactinemia |
| *GnRH agonists* | *Leuprolide acetate* | Subcutaneous | 3.75 mg/30 days 11.25 mg/90 days | **Not Reported** | Male hormones deprivation, flushing |
| | *Triptorelin* | | | | |
| | *Goserelin* | | 3.6 mg/30 days 10.8 mg/90 days | | |
| *Steroidal antiandrogen* | *Spironolactone* | Oral | 100–300 mg/die | **Not Reported** | Hypotension, hyperkalaemia and hyperprolactinemia. |
| *5-α-reductase inhibitor* | *Finasteride* | Oral | 2.5–5 mg/die | **Not Reported** | Depression and sexual dysfunctions |

Before starting the hormonal treatment, clinicians should evaluate possible risk factors for thromboembolic events such as personal history of deep vein thrombosis or modifiable factors (e.g., obesity or smoking). The presence of pathological conditions that could be worsened by estrogen therapy (e.g., hyperprolactinemia, hypertriglyceridemia, cerebrovascular disease) [44] should be also assessed. Furthermore, fertility preservation techniques (e.g., sperm cryopreservation, surgical sperm extraction, and testicular tissue cryopreservation) should be discussed before starting GAHT [9].

Our clinical and therapeutic approach in treating transgender women is depicted in Figure 3.

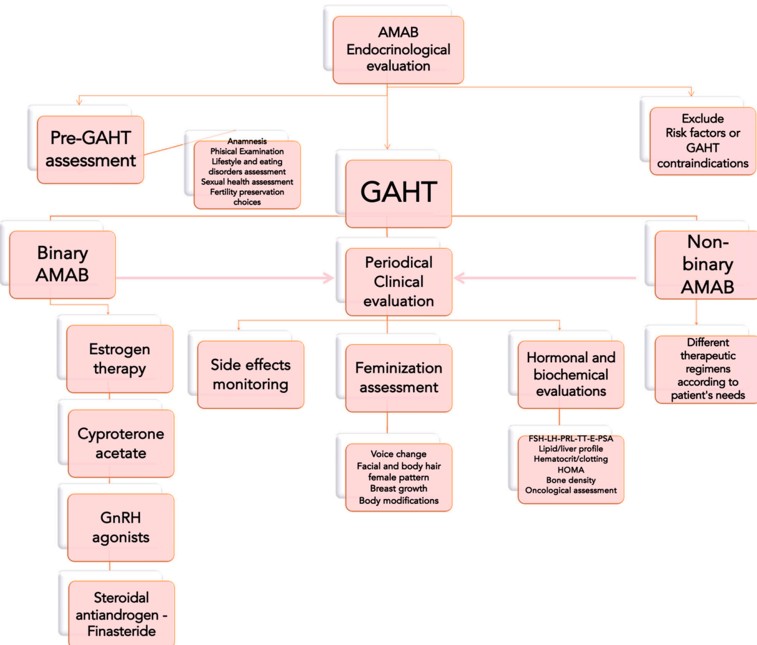

**Figure 3.** Clinical approach to individuals assigned as male at birth (AMAB) and affected by Gender Dysphoria, before and after a Gender Affirming Hormonal Treatment (GAHT). Legends: FSH = Follicle-Stimulating Hormone; LH = Luteinizing Hormone; PRL = Prolactin; TT = total testosterone; E = estradiol; HOMA = Homeostasis Model Assessment for insulin resistance (index).

### 3.1. GAHT Regimens and Side Effects

Regarding estrogen preparations, current guidelines suggest not to use ethynilestradiol in any GAHT regimen due to the potential risk of thromboembolism (1 to 5% incidence) [44]. Transdermal and parental routes seem to be less thrombogenic, even if they are more difficult to monitor [6]. In our clinical practice, we generally prefer to use oral estradiol valerate instead of transdermal or parenteral formulations to minimize their long-term side-effects represented by bone loss, lipid abnormalities, liver alteration, and increased breast/prostate cancer risk.

In almost every transgender woman, estrogen cannot successfully suppress serum total testosterone concentrations to feminine range (<50 ng/dL), so there is a need to also employ a drugs association. To this end, GnRH analogues, progestins with antiandrogen activity, spironolactone, cyproterone acetate, and 5$\alpha$-reductase inhibitors [6], could be successfully employed. GnRH analogues are effective and few side effects have been reported [45]. However, treatment is not standardized worldwide and is very expensive. In non-binary AMAB, GAHT goals could be adjusted according to patients' needs. Thus, different degrees of feminization and demasculinization could be obtained with variable dosages of estrogens and/or employing other drugs (i.e., 5$\alpha$-reductase inhibitors) [18].

No head-to-head trials between testosterone-lowering drugs are available, so local practice and costs are the main guidelines to the choice of the drugs. In fact, many European countries prefer cyproterone acetate combined with oral estrogens while in the United Kingdom GnRH agonists [46] are fully dispensed by the National Health Service (NHS).

After reassignment surgery, testosterone-lowering drugs could be discontinued [46].

There is no agreement about what 5$\alpha$-reductase inhibitors could be used against hair loss. Finasteride seems to be effective without significant adverse events although its use is limited because of the side effects (e.g., depression or persistent sexual dysfunctions) reported in cisgender men and it is not considered a first-line drug [44]. Some patients ask for progestogens to enhance breast growth, but their effectiveness remains poorly documented. Furthermore, many studies in cisgender women suggest that there may be an association between progestin used with estrogen and an increased risk of thromboembolism [46].

It is necessary to avoid high hormone dosages that can lead to hypertension (even if no significant evidences are reported [23]), hepatic dysfunction (often determined by viral hepatitis or infections [46]), and increased thromboembolic risk [6]. In addition, adequate levels are necessary to prevent hot flushes, mood disorders, and osteoporosis [47]. There is no certainty about the threshold value at which side effects are minimal and feminization is maintained [46].

GAHT might induce a reduction in sexual desire, spontaneous erections, and male sexual function within 1–3 months of treatment in non-surgically treated individuals. Nevertheless, only few studies investigated the effects of GAHT on sexual health, desire, and arousal in AMAB transgender individuals, and scientific evidence is still poor and controversial [9].

### 3.2. Clinical, Hormonal, and Biochemical Evaluations

In transgender women during the GAHT period, clinical, hormonal, and biochemical tests should be performed to assess physical changes and to evaluate any therapy-related side effects. In particular, increased BMI with lean mass reduction [48], breast growth (which seems to be complete in 2–3 years), improvement in self-perceived feminine quality of voice [44], and a reduction in Ferriman–Gallwey score and in sebum production [49] should be mentioned.

Serum estradiol and total testosterone should be dosed every three months during the first year of therapy and then once a year. The aim is to maintain premenopausal women levels both for estradiol (100–200 pg/L) and total testosterone (<50 ng/dL) [6]; estradiol peak should not exceed 400 pg/mL [46].

During the treatment with spironolactone, potassium should be measured every three months during the first two years and then annually because of the risk of hyperkalemia [46].

Other biochemical parameters to monitor include alanine aminotransferase, aspartate aminotransferase, alkaline phosphatase, creatinine, hemoglobin, hematocrit, potassium, and the lipid profile. The lipid profile in transgender women is characterized by an increase in HDL and triglycerides [23]. There is no difference between cisgender male, cisgender female, and transgender female range for alanine aminotransferase, and aspartate aminotransferase, alkaline phosphatase, and creatinine values for transgender women resemble men values. By contrast, LDL, hemoglobin, and hematocrit have ranges similar to those of cisgender women [21].

As for transgender men, insulin responses change after GAHT. In transgender women, there is a worsening in insulin sensitivity and post-OGTT incretin responses due to a decreased lean mass and an increase in fat mass [26].

It is also important to perform periodical evaluation of prolactin since increased levels are reported in high dose/long-term estrogen-treated patients. In many cases, prolactin levels turn down with therapy withdrawal. Especially during cyproterone treatment [44], prolactin should be measured at baseline and, in adults, every two years. If prolactin remains above the normal range, imaging tests should be used [6].

D-dimer level monitoring is not recommended during treatment follow up [6]. So far an increased risk for deep venous thrombosis or cardiovascular events has not been reported [44]. Therefore, as for transgender men, guidelines for general population should be followed [6].

Regarding BMD, a relationship with low estradiol levels and trabecular and cortical bone damage has been described [50]. In transgender women, long-term (i.e., 15 years) estrogen therapy seem to be safe and to preserve BMD [51]. Therefore BMD testing can be considered at baseline or in patients over 60 years if at low risk. Estrogen therapy does not seem to be linked with an increased risk for breast cancer and has a controversial role for prostate cancer [52,53]. However, routine screening is recommended: Gynecological counselling and annual digital rectal examination [6].

### 3.3. Psychological Aspects in Transgender Women during Gender Affirming Hormonal Treatment (GAHT)

The effects of GAHT in transgender women have been evaluated in many studies, especially regarding physical effects, side effects, contraindications, and mortality. It is important to know how GAHT affects mental and cognitive health outcomes [3]. In transgender women, GAHT (before gender affirming surgery) consists in the combination of estrogen with anti-androgens [54].

Gonadal steroids, in particular estrogen, have a pronounced effect on the areas of the brain responsible for mood and cognition [55]. Exogenous administration of estrogens and androgens can have multiple negative effects on mental health, making individuals more prone to develop depression or anxiety [56]. However, as stated by Colizzi et al., GAHT has a positive effect on psychiatric conditions in both transgender women and transgender men [57]. The use of cyproterone acetate has also been related to depression but this appears to be generally transient as it occurs during the first six months of use. In order to lower the dose estrogens and to successfully suppress testosterone levels before surgery, cyproterone acetate may be used. No association has been established between suicide and actual use of cyproterone acetate [56].

Several studies suggest that cross sex hormone treatment is associated with improved quality of life [3].

Even if hormonal treatment has a positive effect on the perceived gender incongruence, this should not be considered always resolutive for subjective wellbeing and quality of life since transgender people may encounter other problems such as discrimination, social isolation, and prejudice [54].

Therefore, patients should be referred for psychiatric counseling and care when suicidal ideation and depression are recognized during medical treatment [46].

In a comparative evaluation, transgender women using hormones have shown less psychopathology than those not in treatment. Prolonged use of hormones may be associated with better psychological adjustment [58].

Transgender women are more prone to experiencing negative emotions than transgender men both before and after GAHT [59]. Even if in animal models testosterone administration was associated with aggressive behavior [60], Defreyne et al. found that testosterone therapy was not associated with either increased aggression in transgender men or decreased aggressive behavior in transgender women with anti-androgen and estrogen therapy [61].

Regarding anxiety, there is a reduction in symptoms among middle-aged and elderly transgender adults (over 50 years of age) undergoing hormone therapy. GAHT also appears to have a positive effect on stress [3]. Indeed, Nguyen et al. measured perceived stress levels using a self-reported perceived stress (PSS) scale. After 12 months of GAHT, transgender individuals (n = 70) had a significantly reduced arousal response to cortisol and perceived stress [3].

GAHT appears to have several effects on emotional functioning. It has been shown to improve mood and behavior in individuals experiencing GD, but there are not enough studies about the effects on aggression and anger. The improvement in mental health seems to be strongly linked to the physical changes induced by GAHT [3]. In a very recent study, participants reported that their feeling of "incongruence" increased as their repulsion towards their body did [62].

In addition, many studies show a statistically significant improvement in quality of life (QoL) in transgender women treated with GAHT due to the higher level of body satisfaction and reduced body awkwardness [3]. During a two-year follow-up study, Fisher et al. found that patients undergoing GAHT had reduced concern with GD, body uneasiness, and depressive symptoms retrospectively compared to not-treated patients [38].

However, more longitudinal studies are needed to further assess the effects of GAHT on behavioral and cognitive domains.

## 4. Pharmacological Issues in Transgender People during Gender Affirming Hormonal Treatment (GAHT)

Although current guidelines provide precise recommendations for gender affirming therapies, they do not include indications on gender-related pharmacology and on creatinine clearance (CrCl) estimation and ideal body weight (IBW) calculations, which are necessary to choose the appropriate medication doses. Moreover, adult transgender people might show comorbidities (e.g., liver diseases, kidney failure, HIV infection, neuropsychiatric diseases), which require particular attention to pharmacological and hormonal therapies. Indeed, a correct pharmacological management is crucial to avoid drug toxicity and/or therapeutic failure [63,64].

Consistent differences have been reported in height, weight, and BMI as well as in non-hormonal laboratory value reference ranges (e.g., serum creatinine, hemoglobin, and uric acid) between women and men. Moreover, there are several gender differences in pharmacokinetics and pharmacodynamics and women are more prone to develop severe adverse reaction to some medications. Due to the gender differences in muscular mass, fat, and plasma volume distribution, it is unclear how GAHT impacts on CrCl, renal clearance, and IBW dose in transgender individuals.

Besides these sex and gender considerations, a careful evaluation of transgender patient's renal function is crucial for the correct GAHT dosing and management, which might sometimes benefit of the pharmacologist consultation. Furthermore, the coexistence of medical conditions such as sexually transmitted infections (STI), human immunodeficiency virus (HIV) infection, metabolic and neuropsychiatric disorders, other pharmacological therapies, and substance abuse should also be considered when evaluating renal function and GAHT dosage.

Transgender people are more vulnerable to STI and HIV infection and epidemiological studies have shown a greater prevalence of HIV and HCV infections in transgender individuals than in cisgender people in the United States. It has been reported a HIV prevalence of 14% among transgender women and 3.2% among transgender men (ages 16–66 years) and an overall prevalence of HCV infection of 3.2% in transgender women and 2.4% in transgender men, respectively [65].

Transgender individuals at risk for HIV might benefit pre-exposure prophylaxis (PrEP) agents, although there are many limits to PrEP use and there is a lack of data regarding its efficacy among transgender people. In particular, possible drug-to-drug interactions (DDIs) between GAHT and PrEP represent a major concern for patients [66]. Nevertheless, it has been reported that estrogen interactions might reduce tenofovir and emtricitabine plasmatic concentrations in transgender women [67,68], but more studies are needed to further assess the existing DDIs between GAHT and PrEP. Contrariwise, available data regarding DDI between PrEP and GAHT in transgender men are poorer. As for estradiol, testosterone dose could be adjusted to maintain therapeutic concentration if PrEP impacts on GAHT [67].

The administration of antiretrovirals (ARVs) or HCV direct acting antivirals (DAAs) in transgender individuals might have pharmacokinetic and/or pharmacodynamic DDIs with GAHT [65].

Theoretically, ARVs should not have clinically significant effects on GAHT concentrations. However, it has been reported that non-nucleoside reverse transcriptase inhibitors may reduce estradiol, cyproterone, $5\alpha$-reductase inhibitors, progestogens, and testosterone plasmatic concentrations; moreover, cobicistat may increase plasma exposure to estradiol, ritonavir may enhance/reduce exposure to estradiol, and both cobicistat and ritonavir should increase exposure to progestogens, cyproterone, testosterone, and $5\alpha$-reductase inhibitors [66]. By contrast, selected ARVs such as unboosted integrase inhibitors, doravirine, or rilpivirine seem to have less impact on GAHT. DAAs may also impact on GAHT thus increasing estradiol concentrations but it is unclear if this interaction is clinically relevant [65]. Therefore, the potential DDIs and overlapping side effects between ARVs, DAAs, and GAHT should be carefully considered by clinicians who should consequently provide drug posology adjustments.

Finally, transgender individuals show increased risk for developing neuropsychiatric disturbances. The interactions between GAHT and neurologic drug represents an important medical challenge. In particular, GAHT regimens may commonly show interactions with antiepileptic drugs [69], while data regarding GAHT interactions with antidepressant/antipsychotics are not available. Similarly, to our knowledge, no interactions between GAHT and antidiabetic, antihypertensive, and antidyslipidemic treatments have been described.

## 5. GAHT and COVID-19 Pandemic

The severe acute respiratory syndrome coronavirus (SARS-CoV-2) infection disease (COVID-19) has particularly burdened on transgender and non-binary individuals due to the exacerbation of socio-economic vulnerabilities, barriers to healthcare, mental diseases, and minority stress already experienced prior the pandemic. The lockdown policies have impeded/lowered the access to medical/surgical healthcare (e.g., GAHT, chest/breast surgery), services (e.g., hair removal), and goods, thus determining transgender people's inability to live according their gender and promoting impaired mental health and QoL [70]. While various studies have reported effects of sex hormones (e.g., estrogens, progesterone, androgens) on COVID-19 in cisgender people, no evidence is available at present in transgender individuals because of the exclusion of the transgender community from COVID-19 demographics data, research studies, and public health surveillance systems [71]. It has been shown that cisgender males are affected by COVID-19 more than women [72] and they more frequently need hospitalization. In light of this epidemiologic data, scientists have hypothesized that androgens could somehow influence disease progression [73].

Early reports have proved that the androgen-responsive transmembrane protease serine 2 (TMPRSS2) and the angiotensin converting enzyme 2 (ACE-2) plays an important role in COVID-19 disease because they are targets to SARS-CoV-2 penetration and infection in host cells [73,74]. In particular, androgen might reduce the expression in different tissues of the ACE-2, which is involved in COVID-19 pathophysiology [75]. Indeed, androgen deprivation therapy seems to reduce COVID-19 infection [73] and dexamethasone (commonly used in COVID-19 treatment) reduces testosterone synthesis [76] contributing to its benefic effects. In addition, androgens reduce the ACE-2 expression in the vasculature [74]. Despite the documented androgen action on COVID-19 disease, the majority of intensive care patients show low testosterone levels [77]. This might be due to the viral impairment of Leydig cells function causing hypogonadism, which in turn is linked to systemic inflammation, altered immune response, and increased cardiovascular risk [75].

By contrast, estradiol may increase the expression/activity of ACE-2 in different tissues [75] and estrogen treatment may reduce the inflammatory reactions and the virus titers in animal models [78]. On the basis of the mentioned studies on hormone influence in COVID-19, we could hypothesize a greater protection from severe COVID-19 in transgender women and an increased risk in transgender men. However, we should consider that immune responses are also influenced by many factors such as genetic, biological, hormonal, and behavioral ones. Moreover, the higher rate of cancer, tobacco use, and other chronic conditions in transgender individuals negatively impact on COVID progression and outcome [71].

It is also important to consider that the COVID-19-induced storm i.e., thrombosis and hypercoagulability, may be worsened by GAHT [79]. As in cisgender people, an appropriate screening should be performed especially in transgender subjects presenting with comorbidities and increased risks. Therefore, lower doses of GAHT, temporary withdrawal, or alternative routes of administration (estradiol patches instead of tablets; testosterone gels instead of intramuscular formulations) should be considered whenever COVID-19 occurs.

## 6. Conclusions

Gender dysphoria should be managed by a multidisciplinary team of well experienced physicians, including endocrinologists, psychiatrists, gynecologists/urologists, and surgeons. As for youths, GAHT in adults should consider psychological issues raised by individuals throughout their life and should be administered after the exclusion of contraindications and/or risk factors, and the discussion of any fertility issue. Moreover, there is a strong relationship between GD, GAHT, and behavioral disturbances. Hormone treatments not only have an impact on phenotypic sexual characteristics, but also on metabolisms, cardiovascular system, bone mineral density, mood, and (sexual) behavior. Further studies are necessary for the evaluation of real side effects prevalence during GAHT in both youths and adults, due to the low number of treated individuals and to the presence of a frequent under-evaluated gender-related pharmacology. Particularly, during the Covid-19 pandemic, sexual steroid hormones use (e.g., androgens/estrogens) may be associated with an increased risk of venous thromboembolism occurrence and should be re-considered either in the presence of an infection or during the follow-up. In addition, we highlight that in transgender athletes treated with prohibited drugs during GAHT (i.e., androgens, spironolactone, and so forth) a therapeutic use exemption (TUE) must be requested to the respective Anti-Doping Organization according to the World Anti-Doping Agency criteria [80,81].

In people treated by GAHT prior to gender affirming surgery, unremoved organs such as prostate in transgender women and breast/uterus/ovaries in transgender men should be also carefully monitored during the course of treatment. Based on guidelines and research evidence, the endocrinologist should tailor treatments and safely administer GAHT after evaluating the cost/benefit ratio for each patient.

**Author Contributions:** Conceptualization, A.A., S.I., and G.I.; formal analysis, A.A. and A.B.; investigation, S.I., M.V., and M.C.Z.; resources, S.I., M.C.Z., and S.I.; data curation, F.S.B. and G.I.; writing—original draft preparation, S.I.; writing—review and editing, A.A., G.I., L.D.L., and A.B.; visualization, S.I.; supervision, A.A. and L.D.L. All authors have read and agreed to the published version of the manuscript.

**Funding:** This research received no external funding.

**Institutional Review Board Statement:** Not Applicable.

**Informed Consent Statement:** Not Applicable.

**Conflicts of Interest:** The authors declare no conflict of interest.

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
