# Peer review of "Endocrine Management of Transgender Adults: A Clinical Approach"

_sexes, doi:10.3390/sexes2010009_

Round 1

Reviewer 1 Report

This review summarizes authors' clinical approach for transgender people seeking hormonal-affirming treatment. My major concerns are related to the use of outdated recommendations and terminology. I encourage authors to improve these aspects, especially if they are involved in transgender care.

Abstract:

  • Sex-reverse” therapy is not an appropriate definition. Transgender people do not seek sex reverting treatment, but a gender affirming one. Also, “sex” should not be used as “gender” synonymous.
  • Again, what does “transgender sex” mean?
  • “Transsexuals” is an outdated term

Introduction

  • [27-28] “The prevalence of gender dysphoria cannot be exactly evaluated since different definitions can be used”. Gender Dysphoria has only one definition, being according to the DSM-5 criteria. According to these criteria, several epidemiological studies have been conducted. For example, Nguyen and colleagues (2018), estimated a prevalence of 0,005-0,014% for AMAB and 0,002-0,003% for AFAB transgender individuals.
  • [31] “gender assigned at birth” instead of “sex assigned at birth”
  • [32] the distress does not arise (or not only) from living in an incongruent gender role, but mainly from an incongruent body
  • [33-34] We shall focus our attention on the endocrine treatment of people affected by Gender Dysphoria (GD) as “a marked incongruence between a person’s gender assigned at birth and gender identity” [2]. The expression “affected by” is possibly stigmatizing; furthermore, I suggest the use of the most recent definition of gender dysphoria/gender incongruence (GD/GI) (T’sjoen and colleagues, 2020)
  • [40-42] “During the diagnostic workup, the endocrinologist is required to investigate on the extent of psychiatric comorbidities, especially affective/anxiety problems [4], depression and suicide risk prevalence that seem to be increased in some patients [5]. When hormonal treatment is considered safe and effective it can be started without further investigations.” Many inaccuracies in these sentences and in Figure 1:
  • The change in the ICD-11 questioned the need of mental health professionals as part of the assessment of trans people wishing gender-affirming medical interventions. Accordingly (see T’Sjoen J Sex Med 2020;-:1e15), the health care professionals (HCP) working with trans people may not be a mental health professional, but they should have expertise in mental health to identify those whomay require the input of a mental health professional.
  • There is an apparent incongruence between the text and the proposed algorithm. Indeed, according to the literature (SOC 2012, T’sjoen 2020) the co-existing psychiatric conditions, possibly interfering with the diagnosis of GI/GD or the initiation of GAHT, should be addressed by a Mental Health Professional. The endocrinologist should rule out any possible medical contraindication to the GAHT. Please specify this aspect in the text.
  • Also, authors should be aware that psychiatric conditions should not be considered as comorbidities (as gender dysphoria/gender incongruence is not a psychiatric morbidity…).
  • how hormonal treatment can be considered effective (or non-effective)?
  • Transgender males – transgender females: “males” and “females” are refereed to sex, whereas “women” and “men” is a gender descriptor. Thus, transgender men and women would be more used instead of transgender males and females, even if AMAB and AFAB trans people should be considered the current recommended terminology

  1. Transgender Males

  • [49] Gender identity disorder (GID), also referred to as transsexualism

GID is an outdated term. According to the most recent version of the DSM (Diagnostic and Statistical Manual of Mental Disorders; 5th ed.). “GID” has been substituted by the term “Gender Dysphoria”.

  • [49] what is normal??
  • [50] illness coexisting to what? Gender dysphoria is not an illness.
  • [58] who are hypogonadal men? Cisgender men?
  • [61] MPA / GnRHa in selected cases (testosterone is effective alone to induce amenorrhea)
  • [11] please update the references (many Belgian studies are focused on this topic, see De Roo et al. )
  • [66] more recent studies have evaluated safety profile of hormonal treatment (see those of the ENIGI group)
  • BMI should be spelled out
  • [78-82] can you better explain the ratio for all these biochemical parameters?
  • [84] reverse sex? Do you mean gender assigned at birth or experienced gender?
  • ]87-88] Regarding lipids in patients who undergone hormonal treatment with testosterone, metanalysis did not provide robust evidence. However, an increase in triglycerides [19] and in total cholesterol [16] has been reported.A recent review of the literature by Maraka and colleagues (2019) describes an increase in LDL-Colesterol and triglycerides in transmen receiving Testosterone treatment, as well as a decrease in HDL-C, as mentioned in the most recent position statement (T’Sjoen 2020).
  • Underwent instead of undergone
  • [101] Individuals’ identity is affected by different factors such as ethnicity, nationality, religion, profession, and age. I believe that the expression is too generic and it might be integrated as follows: “Individuals’ gender identity”
  • [104] In the case of female-to-male (FtM) patients testosterone is prescribed.

The expression “FtM” and “MtF” for transgender people is outdated, since it has been considered scarcely representative of the significant part of non-binary transgender individuals, who don’t identify either in the female or male gender. Therefore, the most recent recommendations (T’Sjoen 2020) suggest the use of “Assigned Female At Birth” or AFAB and “Assigned Male At Birth” or AMAB.

  • [107-108) In addition, recent studies consistently document high prevalence of mental health distress, and substance use and abuse in this population [25]. 
    Many other studies assessed the levels of psychopathology in transgender people: 10.1007/978-3-319-68306-5_12

  • [108-110] In individuals undergoing cross sex hormone therapy there is a higher prevalence of mental health disorders (i.e. disproportionate rates of stressful life events due to their gender incongruence that can lead to major psychiatric disorders such as anxiety or substances abuse) [26].
    The phrase should be better formulated: in fact, it might be suggesting that the gender affirming medical treatment is a risk factor for the development of psychiatric pathology: on the contrary, the GAHT is proven to reduce psychopathology and suicidal risk in transgender people, as described by the Authors later in the paper.  

  • [112] Oppression and social stigma are often experienced by transgenders.
    Please specify what is the meaning of “oppression” in this context. The topic has been studied also by J Abnorm Psychol 2017;126:125-136).

  • [120-124] Hormone therapy seems to improve the quality of life and it also seems to reduce anxiety and social distress [31]. Even though the underlying mechanisms are still debated, untreated patients seem to suffer from a higher degree of stress compared to treated patients [32]. However, whether treatment is associated with a better mental well-being is still under debate [31].
    The psychological outcomes of the initiation of GAHT, and its benefits on body uneasiness and GD have been widely studied, see 10.1210/jc.2016-1276).

  • [125-127] Current studies have shown that in transgender individuals there is also an increased risk for eating disorders and an early diagnosis is critically important [33]. Silverstein & al. reported a higher prevalence of purging or frequent binging in women who reported a gender identity conflict [34],while Ålgars & al. found that most transsexuals experienced disordered eating trying to enhance sexual characteristics of their gender identity [35].
    Please be more specific on the relationship between eating disorders and gender dysphoria. For example, Bandini and colleagues (2013) (10.1111/jsm.12062) highlighted how eating disorders in gender dysphoria are characterized by severe body uneasiness. Please specify if the expression “women who reported a gender identity conflict” is referred to female-assigned at birth individuals or to transgender women.

3) Transgender females

  • [144] Therapeutic regimens are more complex than in transgender men (see table 2).
    This affirmation is arbitrary. In which terms the therapy in AMAB people is more complex? If we think in terms of way of administration, testosterone is often administered through a subcutaneous injection, which can be considered much more complicated than the daily assumption of pills or the use of transdermal gel.

  • [159-160] “Some patients require progestogens to enhance breast growth but their effectiveness remains not well documented.” Maybe some patients ask for ?

  • [193-195] On the contrary, low density lipoprotein, hemoglobin and hematocrit resemble female values [17].
    Please specify the meaning of this phrase: do the Autors mean that those parameters should be maintained within the female range, or do they mean that there is no significant change in those parameters due to therapy with estrogens and antiadrogens?

  • [211-212] In MtF transsexuals, GATH (before the sex reassignment surgery) consists in the combination of estrogen with anti-androgens [48].
    "MtF transsexual" is outdated, as well as "sex reassignment surgery" which should be replaced by "gender affirming surgery".

  • [215-216] Estrogens and androgens can have multiple negative effects on mental health, making individuals more prone to develop depression or anxiety [50].
    Please specify, since the phrase seems to mean that every kind of sex steroid, regardless of type and source (endogen/exogen) has a negative impact on mental health. Also, the source of the statement is very dated.

  • [216-218] However, the study of Colizzi & al. However, as stated by Colizzi et al. GAHT has a positive effect on psychiatric comorbidities in both MtF and FtM patients [51].Please re-formulate the phrase since there is a repetition.

  • [218-222] The use of cyproterone acetate has also been related to depression but this appears to be generally transient as it occurs during the first 6 months of use. In order to lower the dose estrogens and to successfully suppress testosterone levels before surgery, cyproterone acetate may be used. At present, no association has been established between suicide and actual use of cyproterone acetate [50].
    The source is very dated, so it is inappropriate to use the expression “at present”.

  • [225] Even if sex reassignment improves gender dysphoria,

The term “improves” might be substituted by “decreases”.

  • [230] Moreover, the complained neurotic and psychotic disorders were considered to be within the normal limits.
    Please specify: does it mean that the neurotic and psychotic disorders went into regression after the initiation of the hormonal treatment, according to the study?
  • [234-236] However, the study conducted by Defreyne & al. found that testosterone therapy was not associated with either increased aggression in FTMs or decreased aggressive behavior in patients MtF With anti-androgen and estrogen therapy [54].What do the Authors mean by “aggression in FtMs”? Please specify.

  • [245-246] In addition, many studies show a statistically significant improvement in QoL in transgender males treated with GAHT due to the higher level of body satisfaction and reduced body [47].

The last phrase is likely to be incomplete: do the Authors intended to write “body uneasiness”? in that case, prospective studies evaluating body uneasiness changes during hormonal treatment should be quoted (doi: 10.1210/jc.2016-1276. ) Above all, I believe that a review exploring the endocrine treatment of transgender adults, which is meant to be really comprehensive of the most recent literature, should mention the existence and prevalence of gender-non-binary individuals, possibly requesting non-standard gender-affirming hormonal treatments in order to achieve a partial masculinization or feminization. Even if there’s still a lack of standardized hormonal treatment protocols, the topic has been recently addressed (10.3390/jcm9061609).

Author Response

Attached is Reply. Highlighthed in yellow, our point-by-point response.

Reviewer 2 Report

This paper offers an overview of hormonal therapies for transgender individuals. As such it is a resource for some practitioners and individuals who would like some information on hormonal therapies. I find that most of the information in the paper is general and does not offer new research or information. It is also not a review of previous studies or information on hormonal therapies that could be useful in a peer-reviewed publication. Even as a review of general practices, there could be more attention focussed on adolescents or children, a combination of hormonal treatments with other medications such as those for HIV, hypertension, or diabetes. Also, there is an inconsistency between the terms used (transsexual and transgender) -- the term transgender is more in line with the literature in this area.

Author Response

Attached file with highlighted in green authors' response

Round 2

Author Response

Attached please find the file in response to this reviewer.

Reviewer 2 Report

You have greatly improved the originality and uniqueness of the paper. I ask that you clarify early on when you use "we" or "our" as it is in the abstract and needs to explained in the abstract and early on in the introduction that you are basing this on your experience and why that is relevant. Additionally, I ask that you write about the need to explore these issues among transgender youth in the discussion/conclusion area.

Author Response

Attached please find the response to this reviewer.
